# Relationships between Maternal Dietary Patterns and Blood Lipid Levels during Pregnancy: A Prospective Cohort Study in Shanghai, China

**DOI:** 10.3390/ijerph18073701

**Published:** 2021-04-01

**Authors:** Na Wang, Zequn Deng, Liming Wen, Yan Ding, Gengsheng He

**Affiliations:** 1Nursing Department, Obstetrics and Gynaecology Hospital of Fudan University, Shanghai 200090, China; nawang16@fudan.edu.cn; 2Department of Nutrition and Food Hygiene, School of Public Health, Fudan University, Shanghai 200032, China; 17211020137@fudan.edu.cn; 3Key Laboratory of Public Health Safety of Ministry of Education, Fudan University, Shanghai 200032, China; 4School of Public Health, University of Sydney, Sydney 2006, Australia; liming.wen@sydney.edu.au; 5Population Health Research and Evaluation Hub, Sydney Local Health District, Sydney 2050, Australia

**Keywords:** dietary patterns, blood lipid levels, pregnancy, cholesterol

## Abstract

The relationships between maternal dietary patterns and blood lipid profile during pregnancy have not been well understood. We aimed to analyze the dietary patterns of pregnant women and their associations with blood lipid concentrations during pregnancy. A cohort of 1008 Chinese pregnant women were followed from 10–15 weeks of gestation to delivery. Their dietary patterns were identified using a principal component analysis. The relationships between dietary pattern score and maternal blood lipid concentrations were assessed using both multivariate linear regression models and generalized estimating equation (GEE) models. Five different dietary patterns were identified. GEE showed that a high score for the fish-shrimps pattern was associated with higher concentrations of total cholesterol (TC) (β = 0.11), low-density lipoprotein cholesterol (LDL-C) (β = 0.07), and high-density lipoprotein cholesterol (HDL-C) (β = 0.03), with all *p* values < 0.001. In contrast, a high tubers-fruit-vegetables pattern score was associated with lower concentrations of TC (β = −0.12), LDL-C (β = −0.07), and HDL-C (β = −0.03), with all *p* values < 0.001. In addition, dietary protein, carbohydrate, and cholesterol intake significantly contributed to the associations between the fish-shrimps dietary pattern and blood lipid concentrations. Predominant seafood consumption is associated with higher cholesterol concentrations, whereas predominant tuber, fruit, and vegetable consumptions are associated with lower cholesterol concentrations during pregnancy.

## 1. Introduction

Pregnancy is a unique physiological period in which a mother undergoes profound changes in glucose and lipid metabolism to ensure adequate supply of nutrients to the fetus despite intermittent maternal food intake [1]. From the 12th week of gestation, there is a steady increase in circulating concentrations of triacylglycerols, fatty acids, cholesterol, and phospholipids [2]. Importantly, cholesterol is used for steroid synthesis and fatty acids are used for membrane formation or are oxidized, in both the mother and fetus. However, dyslipidemia during pregnancy has been shown to increase the risk of adverse pregnancy outcomes, such as gestational diabetes mellitus (GDM), pre-eclampsia, fetal growth disorders, and preterm birth [3,4,5,6], and further affect the long-term health of offspring with a higher risk of childhood obesity and metabolic disease [7,8].

Although the concentrations of estrogen and progesterone and insulin resistance are thought to be principally responsible for the changes of blood lipid concentrations during pregnancy, other factors such as maternal pre-pregnancy body mass index (BMI) [9,10], gestational weight gain (GWG) [6], and socioeconomic characteristics [11] also have effects. Over the last decade, a variety of studies have shown that nutrients, food, and dietary patterns during pregnancy are associated with perinatal outcomes including GDM, hypertensive disorders of pregnancy, and preterm birth [12,13,14]. However, there is little evidence regarding the effects of diet on maternal blood lipid concentrations, especially during pregnancy. A single study conducted in Brazil showed that pre-pregnancy dietary pattern is associated with blood lipid concentrations throughout pregnancy [11]. However, several studies of non-pregnant populations both in Asian and Western countries have shown associations between dietary pattern and blood lipid profile [15,16,17]. The China Health and Nutrition Survey showed that the traditional southern diet was inversely associated with high-density lipoprotein-cholesterol (HDL-C) concentrations; the snacking dietary pattern was significantly associated with higher triacylglycerol (TG) concentrations; and the Western dietary pattern was positively associated with total cholesterol (TC) and low-density lipoprotein-cholesterol (LDL-C) concentrations in 2468 Chinese women [15]. Thus, it is plausible to hypothesize that diet during pregnancy may have the potential to influence blood lipid profiles among Chinese pregnant women.

Up to date, the relationships between maternal dietary patterns and blood lipid profile during pregnancy have not been well understood. Therefore, the aims of the present study were to (1) investigate whether dietary patterns during mid-pregnancy are associated with blood lipid level during pregnancy, and (2) explore which nutrients or types of food may mediate the relationships between the dietary patterns and blood lipid levels during pregnancy.

## 2. Materials and Methods

### 2.1. Study Design

A prospective cohort study was conducted among pregnant women in Shanghai, China, to assess the association between maternal diet and blood lipid profiles during pregnancy. The study was approved by the Research Committee of the Obstetrics and Gynecology of Hospital Fudan University (Ethics Approval number 2017-74), and written consent was obtained from all participants. Briefly, participants were recruited from the Obstetrics and Gynecology Hospital of Fudan University, a tertiary hospital with the total number of births around 15,000 annually in Shanghai. From October 2018 to March 2019, 1500 pregnant women at 10 to 15 weeks of gestation who attended their first routine antenatal care at the hospital were approached and assessed for eligibility by 2 research nurses. Women were eligible if they: (1) Had registered and attended their first antenatal care for a live singleton neonate at <15 weeks’ gestation; (2) planned to give birth in this hospital; and (3) were able to give written consent to participate. Women with familial hyperlipidemia, hypo/hyperthyroidism, Cushing’s syndrome (or on exogenous steroids), or pre-existing diabetes were excluded from the study. At recruitment, participants’ baseline data about their demographic and lifestyle characteristics, and self-reported medical and reproductive history were collected. The maternal medical and reproductive history were crosschecked with their medical records in the hospital database. A total of 1008 out of 1500 women were recruited in this study and 840, who provided a complete set of dietary data with plausible values for total energy intake, were included in the final principal component analysis (PCA). We excluded those who had implausible values of total energy intake (<500 or >3500 kcal/day) [18]. For further analysis of the relationships between dietary pattern and blood lipid concentration, we included women who had had at least one blood lipid measurement during pregnancy (*n* = 829, 789, and 807 for the first, second, and third trimesters, respectively).

### 2.2. Lipid Analysis

As part of routine antenatal care, fasting blood was drawn for TC, TG, HDL-C, and LDL-C at recruitment (10–15 weeks’ gestation) and 24–28 weeks’ gestation. At 32–36 weeks of gestation, fasting blood for TC, TG, HDL-C, and LDL-C were repeated. Venous blood samples were collected in the morning after an 8–12 h fast from all the participants. Samples were then centrifuged at 3000 rpm for 5 min. Serum were separated and stored at −80 °C until biochemical analysis. All the lipid measurements were performed on an automatic biochemical analyzer (Hitachi 7600, Tokyo, Japan) in the biochemical laboratory of Obstetrics and Gynecology Hospital of Fudan University. TG and TC were measured using a colorimetric method (Wako Chemicals, Odawara, Japan), and HDL-C and LDL-C were measured using a direct assay method (Shanghai Beijia Biochemistry Reagents Co., Ltd., Shanghai, China). The interassay coefficients of variation for all these measurements were <10%.

### 2.3. Dietary Assessment

All the women were required to complete two 3-day food records (FRs) at 12–16 and 22–26 weeks of gestation to assess their dietary intake. At the enrollment visit, clinical research staff trained participants to complete the FRs appropriately. Women were provided with colored photographs of over 200 common food items with different portion sizes to help them estimate their food portion sizes. They were instructed to write down all food and beverages consumed, including lists of all ingredients and their portion sizes (in grams or using common household measures) over 2 weekdays and 1 weekend day. A specifically trained researcher then converted these food portions from common household measures, such as tablespoon, cup, and plate, to grams. The completed 6-day FRs were collected when the women underwent routine oral glucose tolerance testing, and the information was entered into an electronic database. We calculated daily food intake using the mean consumption of each type of food during the 6-day period and derived daily energy and nutrient intakes using the China Food Composition Database [19,20].

For the analysis of dietary patterns, all food items consumed were combined into 15 food and beverage groups based on their similarities in nutrient profile including refined grains, whole grains and pulses, vegetables, fungi and mushrooms, algae, fruits, tubers, dairy products, red meat and animal organs, freshwater fish, marine fish, shrimps and shellfishes, beans and bean products, confectionery, and sugary beverages. PCA, with oblique rotation method, was used to derive the dietary patterns based on women’s food intakes. Since the ultimate goal of any rotation is to obtain some theoretically meaningful factors and the simplest factor structure [21], we chose oblique rotation rather than orthogonal rotation to meet the goal based on the food data in our study. The number of factors retained was based on the scree plot and the eigenvalues and factor interpretability after rotation [21]. In this study, food groups that showed the absolute value of factor loadings >0.3 were considered to be important to the interpretability of each pattern. The collinearity of the factors derived by PCA were tested because the oblique rotation method was used, and the results showed that there was no multicollinearity among these factors. Factor scores for each dietary pattern were calculated by summing the intakes of the food groups weighted by their factor loadings. Each woman received a factor score for each dietary pattern with higher scores indicating greater adherence to that dietary pattern. The mean factor score of all the women under each pattern is zero. Dietary patterns were named according to the food groups that contributed most to the pattern.

### 2.4. Covariates

We collected information regarding maternal age, education level, ethnicity, monthly household income, smoking and passive smoking, alcohol drinking, parity, first-degree family history of diabetes, and self-reported pre-pregnancy weight at recruitment. Maternal heights were measured using a calibrated stadiometer at recruitment. Age (in years) was treated as a continuous variable, except for the descriptive statistic for which we categorized as <25, 25–29, 30–34, or ≥35 years. Educational level was classified into five groups: Middle school or below, high school, college, graduate, and postgraduate or above. Ethnicity was categorized into two groups: Han and Other. Monthly household income was categorized as <¥10,000, ¥10,000–¥30,000, and ≥¥30,000. Smoking, passive smoking, and alcohol drinking 3 months preconception were treated as dichotomized variables (yes/no), as was first-degree family history of diabetes. Parity was categorized as 0, 1, or ≥2. Maternal pre-pregnancy BMI was calculated by dividing pre-pregnancy weight in kilograms by the square of height in meters, and participants were classified as underweight (<18.5 kg/m^2^), normal weight (18.5–23.9 kg/m^2^), overweight (24.0–27.9 kg/m^2^), or obese (≥28.0 kg/m^2^), on the basis of the BMI cut-offs defined for Chinese people [22]. Weight gain was calculated by subtracting self-reported pre-pregnancy weight from weight measured at the follow-up check-up. Gestational week was confirmed by ultrasound examination in the first or early in the second trimester, otherwise the date of the final menstrual period was used. Physical activity level during pregnancy was assessed using the short version of the International Physical Activity Questionnaire (IPAQ) [23] during weeks 22–26, from which MET-min per week was derived [24].

### 2.5. Statistical Analysis

Comparisons between groups were performed using χ^2^ tests for categorical variables and ANOVA or t tests for continuous variables with normal distribution. Mann-Whitney and Kruskal–Wallis tests were used for continuous variables with skewed distributions. Frequencies and percentages were used to describe the distributions of categorical variables. Continuous data were presented as means and standard deviations or medians or interquartile ranges (IQRs). Dietary pattern scores were categorized into quartiles to describe the distribution of maternal blood lipid concentrations, and the nutrients and food intakes by participants under each dietary pattern. Blood lipid fractions (TC, HDL-C, LDL-C, and TG) were considered as dependent variables. Multivariate linear regression models were used to assess the relationships between each dietary pattern and each lipid fraction. We further investigated the contributions of energy-yielding macronutrients and dietary cholesterol to the relationships between dietary patterns and blood lipid levels by individually adding each macronutrient and cholesterol to the original adjusted multivariate linear regression models. Since many women had three gestation (time)-dependent blood lipid values, a generalized estimating equation (GEE) model was used to verify the relationships between dietary pattern scores and blood lipid concentrations throughout pregnancy [25,26].

Three adjusted models were constructed to identify the optimal relationship between the main exposure (dietary patterns) and the outcomes (blood lipid levels). In Model 1, only five dietary pattern scores were included. Model 2 was additionally adjusted for demographic covariates (age, education level, ethnicity, monthly household income, smoking and passive smoking, alcohol drinking, parity, first-degree family history of diabetes, and pre-pregnancy BMI). Model 3 was further adjusted for GWG, gestational weeks, energy intake, and MET min/W. All statistical analyses were performed using SPSS statistics version 22.0 (IBM, Inc., Armonk, NY, USA). All tests of statistical significance were two-sided with *p* < 0.05 considered statistically significant.

## 3. Results

### 3.1. Characteristics of the Study Participants

The characteristics of the participants were presented in Table 1. The mean maternal age was 31.0 ± 3.5 and the mean gestational age at recruitment was 11.1 ± 2.1. More than 90% (923/1008, 91.6%) of women had completed a college degree. The prevalence of overweight and obesity was 13.5% (136/1008), whereas underweight women were accounted for 14.9% (150/1008). The percentage of primipara was 80.0% (806/1008).

### 3.2. Dietary Patterns

Using PCA, five distinct dietary patterns with eigenvalues >1.0 were extracted from the scree plot (Appendix A). The first pattern, tubers-fruit-vegetables, included higher intakes of tubers, fruit, vegetables, dairy products (milk, milk powder, and yogurt), and whole grains and pulses (including coarse cereals, red beans, mung beans, and kidney beans). The second pattern, beans-fungi-algae, had higher intakes of mushrooms and fungi, algae, beans, and bean products (soybeans, and soybean milk). The third pattern, fish-shrimps, included higher intakes of both freshwater and marine fish, shrimps, and shellfishes. The fourth pattern, refined grains-red meat-organs, was characterized by higher consumption of refined grains, red meat, processed meat, animal organs and blood, but fewer whole grains. The fifth pattern, confectionery-sugary beverages, was characterized by higher intakes of confectionery and sugar-sweetened beverages. These five patterns accounted for 47.96% of the total variance (Table 2).

### 3.3. Dietary Patterns in Relation to Maternal Characteristics

The relationships between dietary patterns and maternal characteristics are presented Table 3. We found that the tubers-fruit-vegetables pattern score was higher in non-Han women and in primipara. The beans-fungi-algae score was also higher in non-Han women, in those with an educational level of postgraduate or above, and in those with a higher monthly household income. The refined grains-red meat-organs score was lower in women with an educational level of graduate or above and in primipara. The confectionery-sugary beverages score was higher in women who consumed alcohol and experienced passive smoking 3 months before conception.

### 3.4. Dietary Patterns in Relation to Blood Lipid Profile

Figure 1 shows that maternal blood lipid profile (TC, HDL-C, LDL-C, and TG) changed as pregnancy advanced. We found that the serum concentrations of TC, HDL-C, and LDL-C were inversely associated with the tubers-fruit-vegetables pattern score (Appendix A). Table 4 shows that the tubers-fruit-vegetables score was significantly associated with decreased TC, HDL-C, and LDL-C concentrations in all three trimesters. In contrast, consumption of a fish-shrimps pattern was significantly associated with higher TC, HDL-C, and LDL-C concentrations in all three trimesters. Greater adherence to the confectionery-sugary beverages patterns significantly increased TC, HDL-C, and LDL-C concentrations in the second and third trimesters. However, none of the dietary patterns were related to TG concentration during pregnancy. The association of the fish-shrimps pattern with serum cholesterol concentration disappeared in each trimester after additional adjustment for protein-to-carbohydrate ratio or cholesterol, was no longer significant in the second and third trimester after adjustment for carbohydrate and was disappeared except for HDL-C in the third trimester after additional adjustment for protein intake. (Appendix A).

In addition, we performed GEE to fully exam whether a diet pattern could modify blood lipid profile during whole pregnancy. As shown in Table 5, after adjusting for potential confounders, women with higher scores for the tubers-fruit-vegetables pattern had lower TC, LDL-C, and HDL-C concentrations throughout pregnancy. There were positive associations between fish-shrimps pattern score and TC, LDL-C, and HDL-C concentrations. None of the dietary patterns were related to TG concentration during pregnancy.

## 4. Discussion

In the present study, we have identified five dietary patterns in pregnant Chinese women and examined their relationships with blood lipid profiles. Noticeably, there were several important findings regarding these relationships. Firstly, we found that higher adherence to the fish–shrimps pattern was associated with higher concentrations of TC and LDL-C, but also with a higher concentration of HDL-C throughout pregnancy. In contrast, higher adherence to the tubers-fruit-vegetables pattern was associated with lower concentrations of TC and LDL-C, but also with a lower concentration of HDL-C. Secondly, the study also showed a dietary pattern could modify blood lipid profile, specifically cholesterol but not TG. Moreover, we specifically investigated the association of dietary patterns with blood lipid levels with adjustment for dietary macronutrients intakes, and found that the association between dietary pattern and blood cholesterol concentrations could be at least partially attributed to differences in the intakes of dietary protein, carbohydrate, and cholesterol. These results suggested that a high-protein, high cholesterol, and low-carbohydrate dietary pattern in pregnancy was associated with an elevated blood cholesterol concentrations in pregnant Chinese women.

Lipid metabolism is a complex process involving endogenous synthesis and exogenous food intake, as well as the interaction between lipoprotein receptors and enzymes in the body. It can be influenced by a number of dietary factors, including total energy intake, dietary fiber content [27,28], fat composition (saturated vs. unsaturated fatty acids) [29], carbohydrate intake [30], cholesterol intake [31], and whole grain content [32]. In addition to analyzing the effects of individual nutrients and foods, dietary pattern analysis can be used to assess the dietary habits of the population. This approach takes into account the cumulative or antagonistic effects of dietary components and is now widely used.

There have been several studies of the relationships between dietary pattern and blood lipid concentrations in both pregnant and non-pregnant women. Eshriqui [11] showed that pre-pregnancy dietary patterns were associated with blood lipid levels in pregnant Brazil women, and specifically, the fast food and candies pattern was positively associated with TG concentration. In contrast, we found no relationships between dietary patterns and TG. This difference between the two studies could be explained by the differing time points used for the dietary assessments. TG concentration is likely to be influenced by insulin resistance state, as found among women with pre-pregnancy obesity and GDM [9,33]. Weight reduction improves insulin sensitivity and decreases TG levels. Regular physical exercise reduces plasma TG levels over and above the effect of weight reduction [34,35].

A multi-ethnic cohort study that included people of Chinese, Malay, and Indian ethnicity showed that a “healthy” dietary pattern, characterized by high intakes of whole grains, fruit, dairy, vegetables, and unsaturated cooking oil was inversely associated with total cholesterol, LDL-C, and fasting TG concentrations, and positively associated with HDL-C concentration. However, in the sub-group analysis, the “healthy” pattern was inversely associated with HDL-C in Indians, whereas the converse was true in Chinese and Malays [36]. In our study, the tubers-fruit-vegetables pattern, similar to the “healthy” pattern in this previous study, was inversely associated with HDL-C concentration in Chinese women throughout pregnancy. These discrepancies could be due to biological differences in the metabolism of the different ethnic groups or to the differences in the composition of the “healthy” pattern. For example, in the previous study, higher “healthy” pattern scores were more clearly associated with higher fish and seafood intake in Chinese and Malays than in Indians. Coincidently, the tubers-fruit-vegetables pattern in our study contained fewer fish and seafood, which made it more similar to the diet reported for the Indians. A low intake of fats and oils may increase the risk of inadequate intake of vitamin E and essential fatty acids, which may contribute to the lower HDL-C concentration [37].

Recently, a cross-sectional analysis from the PURE study including 18 countries showed that intake of total fat and each type of fat was associated with higher concentrations of total cholesterol and LDL-C, but also with higher HDL-C and apolipoprotein A1 (ApoA1). Furthermore, it was found that high carbohydrate intake was associated with lower total cholesterol, LDL-C, Apo B, HDL-C, and Apo A1 concentrations [30], which were similar findings to those of the present study. From the distribution of nutrients under fish-shrimps pattern, it was observed that the energy obtained from carbohydrate gradually decreased from Quartile 1 to Quartile 4, whereas that from protein and fat gradually increased. This result was further illustrated by additional adjusting for dietary macronutrients intakes in our multi-linear models. All these underline that the proportion of energy provided by the three macronutrients play an important role in the concentrations of blood cholesterol. In addition, our study suggested that evaluation of these nutrients may improve our understanding of why certain dietary patterns affect blood lipid profile during pregnancy.

The strengths of our study included its prospective design, the use of full datasets of blood lipid profile throughout pregnancy and the availability of many covariates that allowed the study to adjust for a number of potential confounders. Another strength was the method of dietary assessment used in the study, because 3-day food records are one of the most reliable methods available [18]. Third, we not only used multivariate linear regression models to analyze the relationships between dietary pattern and blood lipid profile during each trimester, but also GEE, to more thoroughly assess these relationships throughout pregnancy. In addition, the study focused on dietary patterns, rather than individual dietary components, and the design of investigating the association between dietary pattern and blood lipids with adjustment for dietary macronutrient intakes.

The study also had a number of limitations. First, the data-driven approach of deriving dietary patterns and computing corresponding factor scores could not be easily replicated across studies [21]. Second, the population in our study consisted mostly of Han Chinese, thus we were unable to ascertain whether the associations found in this study could be translated to other ethnic groups. However, because plasma cholesterol concentration is determined by numerous dietary and genetic factors [38], the relative homogeneity of our population advantageously reduced this and other unmeasured confounding factors.

## 5. Conclusions

In conclusion, the study found that dietary patterns during pregnancy could be associated blood lipid profile among Chinese pregnant women. Thus, it is not too late to intervene their diet in order to improve their blood lipid profile. However, to lower TG concentrations, women may need to improve their diet before pregnancy. Further studies are required to clarify the associations between diet and blood lipid concentrations during pregnancy, as well as its long-term impact on cardiovascular health, both in mothers and their offspring.

## Figures and Tables

**Figure 1 ijerph-18-03701-f001:**
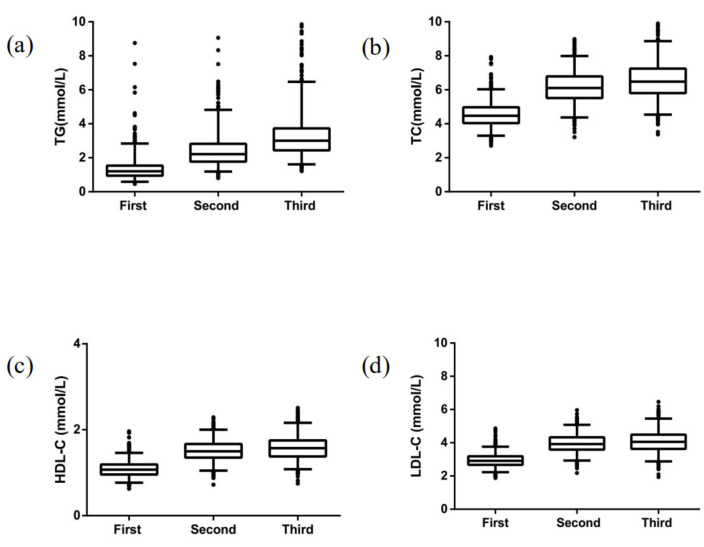
Lipid profile of women during pregnancy across three trimesters (**a**) TG concentrations across three trimesters, (**b**) TC concentrations across three trimesters, (**c**) HDL-C concentrations across three trimesters, (**d**) LDL-C concentration across three trimesters (*n* = 997, 840, and 913 for the first, second, and third trimesters, respectively). TC total cholesterol, HDL-C high-density lipoprotein cholesterol, LDL-C low-density lipoprotein cholesterol, TG triacylglycerol. Data are shown as box and whiskers (2.5–97.5%), non-parametric tests, and the Bonferroni multiple comparison procedure used for the post hoc testing, with all *p* values < 0.001.

**Table 1 ijerph-18-03701-t001:** Characteristics of 1008 study participants in the cohort, Shanghai, China.

Characteristics	Values
Age (years)	
≤29	384 (38.1)
30–34	486 (48.2)
≥35	138 (13.7)
Ethnicity	
Han Chinese	980 (97.2)
Others	28 (2.8)
Education	
≤middle school	21 (2.1)
Senior high school	64 (6.3)
College	205 (20.3)
Graduate,	507 (50.3)
Postgraduate or above	211 (20.9)
Household income	
<¥10,000	81 (8.0)
¥10,000–30,000	649 (64.4)
>¥30,000	278 (27.6)
Smoking 3 months preconception *	
No	984 (97.6)
Yes	24 (2.4)
Passive smoking 3 months preconception ^‡^	
No	892 (88.5)
Yes	116 (11.5)
Alcohol drinking 3 months preconception ^†^	
No	915 (90.8)
Yes	93 (9.2)
Pre-BMI(kg/m^2^)	
<18.5	150 (14.9)
18.5–23.9	722 (71.6)
24.0–27.9	102 (10.1)
≥28.0	34 (3.4)
First-degree family history of diabetes	
No	883 (87.6)
Yes	122 (12.1)
Not clear	3 (0.3)
Parity	
0	806 (80.0)
≥1	202 (20.0)

* Smoking including intermittently or continuously smoking, “no” means zero. ‡ Passive smoking means over 15 min daily exposure to cigarette smoke. † Alcohol intake referring to taking the minimal amount of 40 mL white wine, 125 mL red wine, or 250 mL beer in one month. BMI, body mass index.

**Table 2 ijerph-18-03701-t002:** Factor loading matrix of food groups for the five major dietary patterns identified by Principal Component Analysis (PCA) based on two 3-d food records in pregnant women, Shanghai, China *.

Food Groups	Dietary Patterns				
	Tubers-Fruit-Vegetables	Beans-Fungi-Algae	Fish-Shrimps	Refined Grains-Red Meat-Organs	Confectionery-Sugared Beverages
Tubers	0.690	-	-	-	-
Fruits	0.647	-	-	-	-
Vegetables	0.516	-	-	-	-
Dairy products ^‡^	0.489	-	-	-	-
Fungi and Mushrooms	-	0.653	-	-	-
Beans and bean products ^†^	-	0.645	-	-	-
algae	-	0.539			
Shrimps and shellfishes	-	-	0.648	-	-
Marine fish	-	-	0.552	-	-
Freshwater fishes	-	-	0.541		
Refined grains	-	-	-	0.728	-
Red meat, animal organs and blood	-	-	-	0.568	-
Whole grains and pulses ^Δ^	0.423	-	-	−0.560	
Confectionery	-	-	-	-	0.754
Sugared beverages	-	-	-	-	0.677
Variance explained	15.580	9.098	8.581	7.877	6.822
Cumulative variance explained (%) ^§^	15.580	24.678	33.259	41.136	47.958

* Values are factor loadings (correlation coefficients) between intake of the food groups and the dietary pattern (the factor) derived from principal component analysis. Food groups are sorted by size of loading coefficients. Absolute values <0.30 were not listed for simplicity. ‡ Milk, milk powder, and yogurt. † Soybeans, soybean milk, bean curd, etc. Δ Whole grains and pulses including coarse cereals, red beans, mung beans, and kidney beans. § Percentage of variance in total food intake explained by patterns.

**Table 3 ijerph-18-03701-t003:** Dietary pattern scores, stratified by maternal characteristics, in 840 pregnant women in Shanghai, China.

Characteristics	Dietary Patterns Scores				
	Tubers-Fruit-Vegetables	Beans-Fungi-Algae	Fish-Shrimps	Refined Grains-Red Meat-Organs	Confectionery-Sugared Beverages
Age (years)					
<25		0.15 ± 1.24	0.12 ± 1.23	0.03 ± 1.05	0.04 ± 1.14
25–29	−0.01 ± 0.95	−0.03 ± 0.95	−0.01 ± 0.97	0.00 ± 0.98	−0.02 ± 1.00
30–34	−0.13 ± 0.94	−0.07 ± 0.90	−0.01 ± 0.82	0.06 ± 1.03	0.02 ± 0.85
≥35	−0.17 ± 0.58	0.22 ± 1.14	−0.24 ± 0.76	−0.42 ± 1.11	0.08 ± 0.75
*p* value	0.102	0.187	0.363	0.238	0.906
Ethnicity					
Han Chinese	−0.02 ± 0.99	−0.01 ± 0.98	0.00 ± 0.98	0.00 ± 0.99	0.00 ± 0.97
Others	0.56 ± 1.16	0.48 ± 1.59	−0.14 ± 1.64	−0.09 ± 1.20	0.10 ± 1.81
*p* value	0.007	0.021	0.514	0.661	0.633
Education					
≤Middle school	−0.13 ± 0.57	−0.14 ± 0.59	−0.03 ± 0.90	0.17 ± 0.83	−0.25 ± 0.90
Senior high school	0.00 ± 0.93	0.02 ± 1.60	0.09 ± 1.28	0.43 ± 1.24	−0.25 ± 1.39
College	−0.09 ± 0.93	−0.19 ± 0.88	−0.12 ± 0.93	0.07 ± 0.96	0.13 ± 1.05
Graduate,	0.00 ± 1.11	0.03 ± 0.98	0.08 ± 1.03	−0.04 ± 1.00	0.03 ± 0.96
≥Postgraduate	0.10 ± 0.82	0.12 ± 0.94	−0.11 ± 0.90	−0.10 ± 0.93	−0.10 ± 0.90
*p* value	0.508	0.048	0.087	0.009	0.060
Household income					
<¥10,000	0.01 ± 1.27	−0.16 ± 0.91	0.07 ± 1.05	0.11 ± 0.85	−0.03 ± 1.00
¥10,000–30,000	−0.02 ± 0.92	−0.05 ± 0.96	−0.03 ± 0.94	0.01 ± 1.05	0.00 ± 0.94
>¥30,000	0.04 ± 1.11	0.17 ± 1.11	0.05 ± 1.12	−0.05 ± 0.91	0.02 ± 1.15
*p* value	0.761	0.012	0.556	0.489	0.94
Smoking 3 months preconception *					
No	0.00 ± 1.00	0.00 ± 0.98	−0.01 ± 0.97	−0.01 ± 1.00	−0.01 ± 0.96
Yes	0.16 ± 1.15	0.09 ± 1.70	0.27 ± 1.86	0.41 ± 1.11	0.30 ± 2.03
*p* value	0.472	0.695	0.223	0.065	0.172
Passive smoking 3 months preconception ^‡^					
No	0.02 ± 1.01	0.00 ± 1.02	−0.01 ± 0.99	−0.01 ± 1.01	−0.04 ± 0.99
Yes	−0.18 ± 0.94	−0.01 ± 0.87	0.07 ± 1.09	0.10 ± 0.92	0.28 ± 1.03
*p* value	0.060	0.952	0.471	0.295	0.004
Alcohol drinking 3 months preconception ^†^					
No	−0.01 ± 1.00	0.02 ± 0.99	0.00 ± 0.99	0.01 ± 0.98	−0.03 ± 0.96
Yes	0.07 ± 1.00	−0.20 ± 1.13	−0.02 ± 1.14	−0.08 ± 1.19	0.31 ± 1.33
*p* value	0.507	0.071	0.873	0.459	0.006
Pre-BMI (kg/m^2^)					
<18.5	0.18 ± 1.27	0.15 ± 1.24	0.12 ± 1.23	0.03 ± 1.05	0.04 ± 1.14
18.5–23.9	−0.01 ± 0.95	−0.03 ± 0.95	−0.01 ± 0.97	0.00 ± 0.98	−0.02 ± 1.00
24.0–27.9	−0.13 ± 0.94	−0.07 ± 0.90	−0.01 ± 0.82	0.06 ± 1.03	0.02 ± 0.85
≥28.0	−0.17 ± 0.58	0.22 ± 1.14	−0.24 ± 0.76	−0.42 ± 1.11	0.08 ± 0.75
*p* value	0.102	0.187	0.363	0.238	0.906
First-degree family history of diabetes					
No	0.01 ± 1.00	0.00 ± 0.99	0.00 ± 1.00	−0.01 ± 0.98	−0.01 ± 1.02
Yes	−0.04 ± 1.00	0.03 ± 1.10	0.02 ± 1.00	0.05 ± 1.14	0.05 ± 0.88
Not clear	−0.91 ± 0.15	−0.60 ± 0.32	0.34 ± 0.45	0.20 ± 0.34	−0.06 ± 0.25
*p* value	0.260	0.556	0.825	0.812	0.885
Parity					
0	0.04 ± 1.04	0.02 ± 1.03	0.02 ± 1.00	−0.05 ± 1.01	0.02 ± 1.04
1	−0.18 ± 0.75	−0.07 ± 0.86	−0.10 ± 1.02	0.22 ± 0.95	−0.11 ± 0.82
2	−0.68 ± 0.59	−0.57 ± 0.93	−0.31 ± 0.37	0.18 ± 0.83	0.16 ± 0.03
*p* value	0.031	0.427	0.362	0.013	0.351

* Smoking including intermittently or continuously smoking, “no” means zero. ‡ Passive smoking means over 15 min daily exposure to cigarette smoke. † Alcohol intake referring to taking the minimal amount of 40 mL white wine, 125 mL red wine, or 250 mL beer in one month. BMI, body mass index.

**Table 4 ijerph-18-03701-t004:** The associations between dietary pattern scores and blood lipid levels during pregnancy *^,‡^.

Outcomes	Tubers-Fruit-Vegetables	*p* Value	Beans-Fungi-Algae	*p* Value	Fish-Shrimps	*p* Value	Refined Grains-Red Meat-Organs	*p* Value	Confectionery-Sugared Beverages	*p* Value
	β (95%CI)		β (95%CI)		β (95%CI)		β (95%CI)		β (95%CI)	
First trimester										
TC	−0.09 (−0.14, −0.04)	0.001	−0.02 (−0.07, 0.03)	0.481	0.09 (0.04, 0.14)	0.001	−0.03 (−0.08, 0.02)	0.238	0.03 (−0.03, 0.08)	0.351
HDL-C	−0.02 (−0.04, −0.01)	<0.001	−0.01 (−0.02, 0.01)	0.436	0.02 (0.01, 0.04)	<0.001	−0.01 (−0.02, 0.01)	0.276	0.01 (−0.01,0.02)	0.209
LDL-C	−0.05 (−0.08, −0.02)	0.001	−0.02 (−0.05, 0.01)	0.170	0.04 (0.01, 0.07)	0.012	−0.01 (−0.04, 0.02)	0.617	0.01 (−0.02, 0.04)	0.501
TG	−0.02 (−0.06, 0.03)	0.551	0.03 (−0.02, 0.08)	0.200	−0.02 (−0.07, 0.03)	0.354	−0.02 (−0.06, 0.03)	0.539	−0.02 (−0.07, 0.03)	0.393
Second trimester										
TC	−0.15 (−0.22, −0.07)	<0.001	−0.04 (−0.11, 0.03)	0.311	0.11 (0.04, 0.18)	0.003	−0.03 (−0.10, 0.05)	0.501	0.08 (0.01, 0.16)	0.028
HDL-C	−0.03 (−0.05, −0.01)	0.001	−0.01 (−0.03, 0.01)	0.164	0.03 (0.01, 0.05)	0.001	−0.01 (−0.03, 0.01)	0.575	0.02 (0.01, 0.04)	0.015
LDL-C	−0.08 (−0.12, −0.03)	0.001	−0.02 (−0.07, 0.02)	0.259	0.07 (0.02, 0.11)	0.002	−0.01 (−0.05, 0.04)	0.740	0.05 (0.00, 0.09)	0.031
TG	−0.03 (−0.11, 0.04)	0.368	0.03 (−0.04, 0.10)	0.360	−0.04 (−0.11, 0.03)	0.292	0.03 (−0.05, 0.10)	0.454	0.02 (−0.05, 0.10)	0.510
Third trimester										
TC	−0.17 (−0.25, −0.08)	<0.001	−0.03 (−0.11, 0.05)	0.476	0.14 (0.06, 0.23)	0.001	−0.05 (−0.14, 0.04)	0.242	0.12 (0.03, 0.20)	0.011
HDL-C	−0.05 (−0.07, −0.03)	<0.001	0.00 (−0.03, 0.02)	0.697	0.04 (0.02, 0.06)	<0.001	−0.01 (−0.04, 0.01)	0.215	0.03 (0.01, 0.05)	0.013
LDL-C	−0.10 (−0.16, −0.05)	<0.001	−0.01 (−0.07, 0.04)	0.572	0.09 (0.03, 0.14)	0.001	−0.04 (−0.09, 0.02)	0.173	0.07 (0.02, 0.12)	0.012
TG	−0.06 (−0.15, 0.04)	0.250	0.04 (−0.05, 0.13)	0.428	−0.05 (−0.14, 0.05)	0.340	−0.04 (−0.14, 0.05)	0.391	0.02 (−0.08, 0.11)	0.703

* Multivariate linear models were adjusted for maternal age, Pre-BMI, ethnology, parity, education, household income, smoking and passive smoking, alcohol drinking, pre-pregnancy BMI, and first-degree family history of diabetes. MET-min/W, other dietary patterns, energy intake, gestational weight gain (GWG), and gestational weeks. ‡ The daily food intakes were calculated by averaging the dietary intake of the two time points (12–16 weeks and 22–26 weeks). TC: Total cholesterol, HDL-C: High-density lipoprotein cholesterol, LDL-C: Low-density lipoprotein cholesterol, TG: Triacylglycerol

**Table 5 ijerph-18-03701-t005:** Longitudinal associations between dietary patterns and serum lipid concentrations during pregnancy *.

Outcome	Tubers-Fruit-Vegetables	*p* Value	Beans-Fungi-Algae	*p* Value	Fish-Shrimps	*p* Value	Refined Grains-Red Meat-Organs	*p* Value	Confectionery-Sugared Beverages	*p* Value
	b (95%CI)		b (95%CI)		b (95%CI)		b (95%CI)		b (95%CI)	
Model 1										
TC	−0.14 (−0.20, −0.08)	<0.001	−0.03 (−0.08, 0.03)	0.330	0.10 (0.04, 0.15)	0.001	−0.02 (−0.07, 0.04)	0.497	0.07 (0.01, 0.13)	0.025
HDL-C	−0.04 (−0.05, −0.02)	<0.001	−0.01 (−0.02, 0.01)	0.353	0.03 (0.01, 0.04)	<0.001	0.00 (−0.02, 0.01)	0.565	0.02 (0.00, 0.03)	0.013
LDL-C	−0.08 (−0.11, −0.05)	<0.001	−0.02 (−0.05, 0.02)	0.362	0.06 (0.03, 0.09)	<0.001	−0.01 (−0.04, 0.02)	0.605	0.04 (0.01, 0.08)	0.016
TG	−0.04 (−0.10, 0.03)	0.257	0.02 (−0.05, 0.09)	0.520	−0.03 (−0.09, 0.02)	0.247	0.01 (−0.05, 0.06)	0.803	0.01 (−0.05, 0.07)	0.648
Model 2										
TC	−0.14 (−0.19, −0.08)	<0.001	−0.03 (−0.08, 0.03)	0.344	0.10 (0.04,0.15)	0.001	−0.02 (−0.07, 0.04)	0.504	0.06 (0.00, 0.12)	0.055
HDL-C	−0.04 (−0.05, −0.02)	<0.001	−0.01 (−0.02, 0.01)	0.371	0.03 (0.01, 0.04)	<0.001	0.00 (−0.02, 0.01)	0.574	0.02 (0.00, 0.03)	0.036
LDL-C	−0.08 (−0.11, −0.04)	<0.001	−0.02 (−0.05, 0.02)	0.343	0.06 (0.03,0.09)	<0.001	−0.01 (−0.04, 0.02)	0.573	0.04 (0.00, 0.07)	0.041
TG	−0.02 (−0.08, 0.03)	0.419	0.04 (−0.03, 0.11)	0.309	−0.03 (−0.09, 0.02)	0.242	0.01 (−0.04, 0.07)	0.676	0.02 (−0.04, 0.07)	0.587
Model 3										
TC	−0.12 (−0.18, −0.06)	<0.001	−0.02 (−0.08, 0.03)	0.447	0.11(0.05, 0.17)	<0.001	−0.04 (−0.10, 0.03)	0.251	0.06 (0.00, 0.12)	0.065
HDL-C	−0.03 (−0.05, −0.02)	<0.001	0.00 (−0.02, 0.01)	0.568	0.03 (0.02, 0.04)	<0.001	−0.01(−0.02, 0.01)	0.241	0.01 (0.00, 0.03)	0.060
LDL-C	−0.07 (−0.11, −0.04)	<0.001	−0.01(−0.04, 0.02)	0.489	0.07 (0.03, 0.10)	<0.001	−0.02 (−0.06, 0.01)	0.211	0.03 (0.00, 0.07)	0.073
TG	−0.03 (−0.09, 0.03)	0.285	0.03(−0.03, 0.10)	0.331	−0.04 (−0.09, 0.02)	0.232	−0.03 (−0.08, 0.03)	0.379	0.00 (−0.06, 0.06)	0.964

* Generalized estimating equations model. Model 1: Adjusted for other dietary patterns. Model 2: Model 1 + maternal age, Pre-BMI, ethnology, parity, education, household income, smoking and passive smoking, alcohol drinking, pre-pregnancy BMI, and first-degree family history of diabetes. Model 3: Model 2 + MET-min/W, energy intake, GWG, and gestational weeks. TC: Total cholesterol, HDL-C: High-density lipoprotein cholesterol, LDL-C: Low-density lipoprotein cholesterol, TG: Triacylglycerol.

## Data Availability

The original data supporting reported results was generated during the study. Data supporting reported results are available on request from the study team.

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
