# Peer review of "Relationships between Maternal Dietary Patterns and Blood Lipid Levels during Pregnancy: A Prospective Cohort Study in Shanghai, China"

_ijerph, 2021, doi:10.3390/ijerph18073701_

Round 1
Reviewer 1 Report
1/ there is missing a table with the description of analysed subjects - characterisctics collected are summarised under 3.1. but the data are not presented
2/ Informations about the numbers of subjects within the 5 dietary patterns is also mising. Which is the most common one, is it in agreement with the general patterns observed in the population? Have pregnant females changed theirs dietary habits as a consequence of the pregnancy, or did they continued with the pattern before?
3/ table 2 needs to be better formated, it is difficult to read.
4/ informations from table 3 will profit from the conversion into the figures. additionally, most of the values from the table are repeated within the text, which is redundant.
Reviewer 2 Report
Thank you for the opportunity to review this manuscript by Wang et al. This interesting study used data from a prospective cohort study to examine the trimester-specific and longitudinal associations between empirically derived maternal dietary patterns and blood lipid levels during pregnancy in Shanghai, China. The study reports that predominant seafood consumption is associated with higher cholesterol concentrations, and that predominant tuber, fruit, and vegetable consumptions are associated with lower cholesterol concentrations. The manuscript is very well-written, and the statistical approach is clearly described. Please find below my questions, comments, and suggestions for your consideration for the further improvement of the manuscript.
- Line 24: Abstract: “…with all P values <0.001." I suggest rewording to clarify whether this statement only applies to the tubers-fruit-vegetable pattern or also the fish-shrimps pattern.
- Line 34: “During 12 weeks of gestation…” I wonder whether it should be “During THE FIRST 12 weeks of gestation…” (i.e., approximately during the first trimester).
- The Introduction section provides a clear case of why maternal dietary patterns during pregnancy may be associated with blood lipid levels. To make the rationale of the study even more compelling, I would suggest providing information on the burden of abnormal blood lipids (e.g., elevated TC, TG, and LDL-C, or low HDL-C) among pregnant women in China, is any population-level statistics are available from population registries or prior literature.
- Dietary intake: based on information provided in lines 117-118, the dietary intake was used as the average of the two time points (12-16 wks and 22-26 wks) when diet records were collected. Please clarify whether this approach of averaging the two measures was applied for all analyses, especially in the non-longitudinal, trimester-specific analyses in Table 3. For example, the second diet record was collected after the first-trimester blood lipids. Therefore, for the non-longitudinal analysis of the first-trimester outcomes, I wonder whether there may be any concerns of using the average intake regarding temporality since part of the exposure occurs after the outcome.
- Related to the previous point, if the first-trimester and second-trimester diet records were averaged and used as the exposure to derive dietary patterns, whether there would be any concern regarding the change of dietary intake from first to second trimester that may potentially be masked by taking the average.
- Lines 196-197: “Using PCA, five distinct dietary patterns with eigenvalues > 1.0, as well as factor loadings >0.3, were extracted from the scree plot.” The eigenvalue is specific to each dietary pattern, but factor loading is specific to each individual food group and is not directly extractable from the scree plot. The authors may be referring to the practice of only showing the factor loadings for food groups with large factor loadings (not from the scree plot, but the exact table of factor loadings). Also, I suggest rewording it to be "the absolute value" of factor loading as the loading can be in two directions (positive or negative).
- Table 1: I suggest adding an additional row to show the variance independently explained by each pattern (even though this information is implicit and can be calculated using the cumulative variance explained).
- It is often customary to include the table of basic characteristics in the main text (i.e., as the typical “Table 1”). I wonder whether the authors may consider moving that table from the supplement to be the first main table.
- I may have missed it, but Supplemental Table 2 does not appear to have been cited in the main text.
- As an observational study, I think the potential of residual confounding may also need to be acknowledged as an additional limitation. For example, the family history of dyslipidemia may affect both dietary intake and blood lipids and may confer residual confounding if not measured and adjusted for.
Reviewer 3 Report
The authors describe a well-designed study with important findings. The manuscript contains thorough descriptions of the methods and analyses.
Comments:
Supplemental table 1 and Figure 1: include number of participants in the title.
Figure 1: You state that all p values were <0.001. Clarify whether each trimester was different from the other two for each panel.
How were the questions regarding smoking and alcohol intake phrased? Did "no" mean zero drinks and cigarettes or were zero and minimal amounts combined?
"Whole grains" do not normally include legumes/pulses. Consider modifying food group name to "Whole grains and pulses."
Line 335: "...an important role..."
Conclusion: Before stating that blood lipids profiles can be improved by diet modifications in pregnant women, please describe whether any of the dietary patterns were associated with abnormal/undesirable changes in blood lipid profile. Do data exist showing blood lipid profile and pregnancy outcome?
